# Hydrogen Production Improvement on Water Decomposition Through Internal Interfacial Charge Transfer in $M_3(PO_4)_2$-$M_2P_2O_7$ Mixed-Phase Catalyst (M = Co, Ni, and Cu)

**Junyeong Kim, Jun Neoung Heo, Jeong Yeon Do \*, Seog Joon Yoon, Youngsoo Kim** and **Misook Kang \***

Department of Chemistry, College of Science, Yeungnam University, Gyeongsan, Gyeongbuk 38541, Korea
* Correspondence: daengi77@ynu.ac.kr (J.Y.D.); mskang@ynu.ac.kr (M.K.);
    Tel.: +82-53-810-3798 (J.Y.D.); +82-53-810-2363 (M.K.); Fax: +82-53-815-5412 (M.K.)

**Abstract:** In this study, three types of Nasicon-type materials, $Co_3(PO_4)_2$-$CO_2P_2O_7$, $Ni_3(PO_4)_2$-$Ni_2P_2O_7$, and $Cu_3(PO_4)_2$-$Cu_2P_2O_7$, were synthesized as mixed-phase catalysts (MPCs) for evaluating their potential as new photocatalytic candidates (called $Co_3(PO_4)_2$-$CO_2P_2O_7$mpc, $Ni_3(PO_4)_2$-$Ni_2P_2O_7$mpc, and $Cu_3(PO_4)_2$-$Cu_2P_2O_7$mpc herein). Based on various physical properties, it was confirmed that there are two phases, $M_3(PO_4)_2$ and $M_2P_2O_7$, in which a similar phase equilibrium energy coexists. These colored powders showed UV and visible light responses suitable to our aim of developing 365-nm light-response photocatalysts for overall water-splitting. The photocatalytic performance of $Ni_2(PO_4)_3$-$Ni_2P_2O_7$ MPC showed negligible or no activity toward $H_2$ evolution. However, $Co_2(PO_4)_3$-$Co_2P_2O_7$ MPC and $Cu_3(PO_4)_2$-$Cu_2P_2O_7$ MPC were determined as interesting materials because of their ability to absorb visible light within a suitable band. Moreover, an internal interface charge transfer was suggested to occur that would lower the recombination rate of electrons and holes. For $Cu_3(PO_4)_2$-$Cu_2P_2O_7$ MPC, the charge separation between the electron and hole was advantageously achieved, a water-splitting reaction was promoted, and hydrogen generation was considerably increased. The performance of a catalyst depended on the nature of the active metal added. In addition, the performance of the catalyst was improved when electrons migrated between the inter-phases despite the lack of a heterojunction with other crystals.

**Keywords:** $M_3(PO_4)_2$-$M_2P_2O_7$ mixed-phase catalyst; hydrogen production; water photosplitting; Internal interfacial charge transfer

## 1. Introduction

Since the hydrogen production performance of a $TiO_2$ photocatalyst was first reported by Honda-Fujishima [1–4], numerous studies have been conducted to improve the performance of photocatalysts using efficient light harvesting. As a result, researchers have concluded that if electrons and holes generated by light can be advantageously separated, their recombination could be delayed to increase the frequency of participation in the oxidation-reduction reaction occurring at the interface between the catalyst and reactant [5,6]. To slow down the recombination of electrons and holes, several methods are available, including a one-step excitation system [7], a photosensitized semiconductor system [8], a two-step excitation semiconductor heterojunction system [9], and a Z-scheme system [10]. Catalysts based on such systems have achieved unexpectedly good results, and many related studies have been published [11–13]. Currently, various photocatalysts ranging from widely used $TiO_2$ particles



to metal-sulfide [14], metal-nitride [15], and metal-tungstate [16] with a slightly smaller bandgap have been studied.

Nasicon-type materials such as metal phosphate particles, which are relatively small with a bandgap of approximately 2–3 eV compared to $TiO_2$, have recently attracted the interest of a few researchers [17,18]. In general, Nasicon is a crystalline solid of $A_1B_2(PO_4)_3$, where A is a monovalent cation and B is a single ion or a combination of tri-, tetra-, and penta-atomic ions. Here, the monovalent A ions can move within the lattice with low activation energy [19]. During the early 1980s, Susman et al. succeeded in synthesizing a compound with a formula of $Na_{1+x}Zr_2Si_xP_{3-x}O_{12}$ $(0 < x < 3)$ [20], and because Na, Zr, and Si ions exhibit unique ion conductivity by substituting other elements, this compound has been widely applied as a secondary battery material [21,22]. To date, many ion-exchanged derivatives and substituted Nasicon frameworks are known, e.g., $AM_2(PO_4)_3$ (A = Na, Ca, Sr, Ba; M = Ti, Mo). These structural-type materials consist of a three-dimensional network made up of $PO_4$ tetrahedra, sharing corners with $MO_6$ octahedra and forming interconnected tunnels, where $M^+$ ($Na^+$, $K^+$, $Ag^+$) or oxygen ions have freedom of movement [23]. During a photocatalytic reaction, the transfer of electrons through the formation of interconnected tunnels or the transfer of oxygen atoms can increase the adsorption of the reactants, effectively isolating the photo-induced charge, and consequently contributing to an improved photoactivity [24]. As examples, Fu, et al. reported the photocatalytic activity of $MgTi_4(PO_4)_6$ and $CaTi_4(PO_4)_6$ glass-ceramics containing Nasicon-type crystals [25], and Palla et al. suggested the photocatalytic degradation of organic dyes with $Sn^{2+}$- and $Ag^+$-substituted $K_3Nb_3WO_9(PO_4)_2$ under visible light irradiation [26]. However, this remains in the early stage of research, and has thus, not shown a remarkable catalytic activity. Thus, in this study, $M_3(PO_4)_2$ particles as Nasicon-type materials were prepared by binding phosphate ions to a Co, Ni, or Cu metal in a divalent oxidation state with 7, 8, and 9 electrons in a 3d-orbit of the transition metal. These particles were used as a water-splitting catalyst to compare their hydrogen production performance.

## 2. Results and Discussion

*Characteristics of $M_3(PO_4)_2$-$M_2P_2O_7$ Mixed-Phase Catalysts*

Figure 1 shows X-ray diffraction (XRD) patterns of the as-synthesized catalysts. A sample of cobalt phosphate was shown to have a consistent pattern with the crystal structure of $Co_3(PO_4)_2$ in a monoclinic crystal system [27], and a small XRD peak, belonging to the cobalt phosphate hydrate (monoclinic, P21/n), was also found at 33°. Nickel phosphate particles mostly show an XRD pattern of $Ni_3(PO_4)_2$ in a monoclinic crystal system with a space group of P21/a, although the $Ni_2P_2O_7$ phase of the monoclinic crystal system with the P21/c space group was also mixed [28]. The $Cu_2P_2O_7$ phase (C2/c space group) was also mixed with the $Cu_3(PO_4)_2$ monoclinic crystal (C2/c space group) [29]. From this result, we confirmed that the $M_3(PO_4)_2$ and $M_2P_2O_7$ crystal phases coexist because the energy phase equilibrium of the two crystals lies at approximately the same positions [30]. Jain et al. [31] observed that the mixed-phased $M_x(P_2O_7)(PO_4)_2$ is energetically stable and can be decomposed into $M_xP_2O_7$ and $M_x(PO_4)_3$, although the overall energy of the decomposition is unfavorable by 5 meV/atom. The authors concluded that $M_x(P_2O_7)(PO_4)_2$ has a stable phase. Furthermore, in this study, we did not intentionally attempt to produce two separate hetero-type phases, but the two phases are instead naturally produced during synthesis.

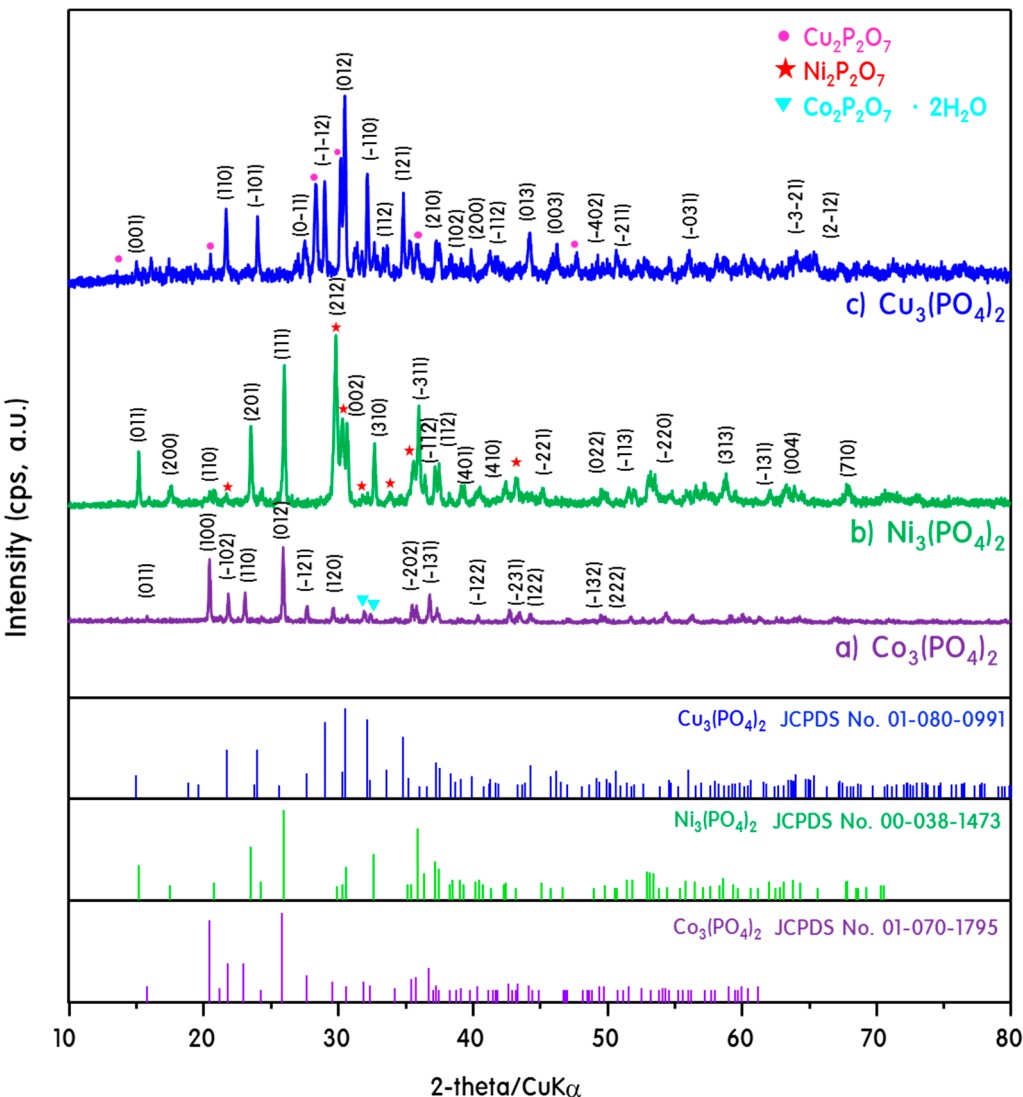

**Figure 1.** X-ray diffraction (XRD) patterns of the $M_3(PO_4)_2$-$M_2P_2O_7$mpc (M = Co, Ni, and Cu).

Transmission electron microscopy (TEM) images are shown in Figure 2. In most studies [32,33], TEM images of $M_2P_2O_7$ are expressed in sheet form. We also interpreted them in the same context. In the photograph showing cobalt phosphate, we can see uniform particles of an elliptical shape with a width of 300 nm and a length of 150 nm. However, it was confirmed in the images that nickel phosphate and copper phosphate were mixed with unclear particles: $Ni_3(PO_4)_2$ with small particles of 50–100 nm was clustered, and a wide sheet of $Ni_2P_2O_7$ particles could also be seen. In the copper phosphate image, it was confirmed that rectangular $Cu_3(PO_4)_2$ particles with a width of 500 nm and a height of 300 nm were mixed with sheet-type $Cu_2P_2O_7$ particles. From the XRD and TEM results, we recognized that the synthesized catalysts coexisted as two crystal phases. Herein, we refer to the $M_3(PO_4)_2$-$M_2P_2O_7$ mixed-phase catalysts (MPCs) as $Co_3(PO_4)_2$-$CO_2P_2O_7$mpc, $Ni_3(PO_4)_2$-$Ni_2P_2O_7$mpc, and $Cu_3(PO_4)_2$-$Cu_2P_2O_7$mpc.

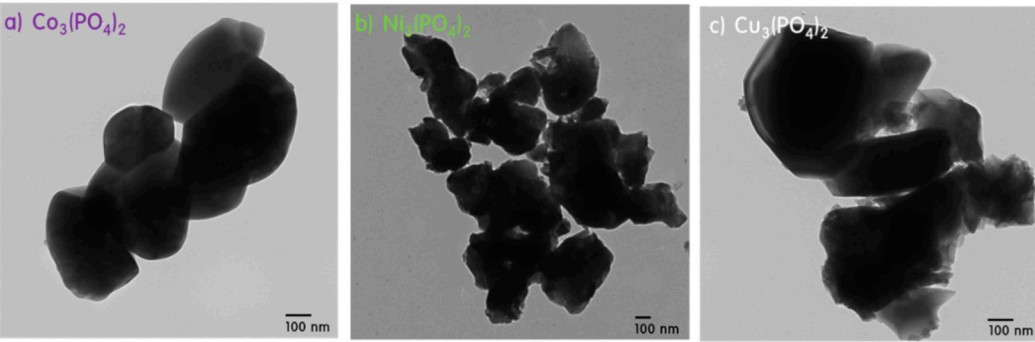

**Figure 2.** Transmission Electron Microscope (TEM) images of the (**a**) $Co_3(PO_4)_2$-$Co_2P_2O_7$mpc, (**b**) $Ni_3(PO_4)_2$mpc, (**c**) $Cu_3(PO_4)_2$-$Cu_2P_2O_7$mpc.

An X-ray photoelectron spectroscopy analysis was conducted on $Cu_3(PO_4)_2$-$Cu_2P_2O_7$mpc as a representative catalyst, the results of which are shown in Figure 3. The peaks at binding energies of 935.7, 133.1 and 530.6 eV refer to the $Cu2p_{3/2}$, $P2p_{3/2}$ and O1s spectra, respectively. All core peaks in $Cu_3(PO_4)_2$-$Cu_2P_2O_7$mpc were recorded, and as shown in Figure 3, the core peak of Cu2p showed two main spin-orbit components at 935.7 and 955.6 eV, corresponding to $Cu2p_{3/2}$ and $Cu2p_{3/2}$, which confirmed the presence of $Cu^{2+}$ [34]. Two peaks located at 942.5 and 962.5 eV are the "shake up" satellite peaks of Cu2p. The high-resolution spectra of P2p show two main spin-orbit components at 133.1 and 134.8 eV, corresponding to $Cu2p_{3/2}$ and $Cu2p_{3/2}$, respectively, thereby confirming the presence of $P^{5+}$. The O1s regions for $Cu_3(PO_4)_2$ located at 530.6 and 532.4 eV are assigned to the M–O–P and P–O–P bonds, respectively [35]. In particular, a peak at 532.5 eV is shown, which was assigned to the P = O of $Cu_2P_2O_7$ [36]. This is similar to the XRD result, and it was concluded that $Cu_3(PO_4)_2$ coexists with the $Cu_2P_2O_7$ phase in $Cu_3(PO_4)_2$-$Cu_2P_2O_7$mpc.

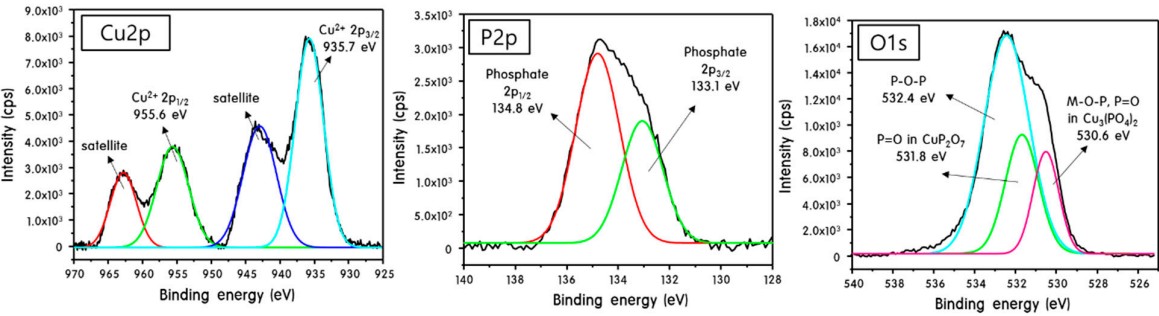

**Figure 3.** XPS spectra of the $Cu_3(PO_4)_2$-$Cu_2P_2O_7$mpc.

Figure 4 shows the DIR-UV-visible absorbance of three $M_3(PO_4)_2$-$M_2P_2O_7$mpc particles. In general, according to a Sugano–Tanabe diagram [37], three absorption curves are allowed in $Co^{2+}$ and $Ni^{2+}$ complexes with $d^7$- and $d^8$-electron configurations. The first curve is observed in the UV region below 300 nm ($T_{1g} \leftrightarrow A_{2g}$), the second curve is shown in the visible region of 400–750 nm ($T_{1g} \leftrightarrow T_{1g}$), and the last curve can be seen in the IR region at above 800 nm ($T_{1g} \leftrightarrow T_{2g}$). In general, within the visible region, the blue $Cu_3(PO_4)_2$-$Cu_2P_2O_7$mpc particles absorb a red wavelength of approximately 650 nm and the yellow $Ni_3(PO_4)_2$-$Ni_2P_2O_7$mpc particles absorb a blue wavelength of 450 nm and a red wavelength within the vicinity of 700 nm. The purple $Co_3(PO_4)_2$-$Co_2P_2O_7$mpc particles absorb a yellow-green wavelength of approximately 550–600 nm. In particular, $Cu_3(PO_4)_2$-$Cu_2P_2O_7$mpc particles show a strong absorption band at above 650 nm. It is known that the diluted $Cu^{2+}$ ions inside a glass matrix exhibit a broad optical absorption of approximately 700 nm, which is assigned to the $^2B_{2g} \rightarrow ^2B_{1g}$ transition owing to the Jahn–Teller splitting of the 3d levels of $Cu^{2+}$ ions in a ligand field [38]. The broad absorption band was observed in the NIR region, which corresponds to three possible d-d electronic absorption transitions for distorted octahedral coordination rather than perfect

octahedral coordination. Thus, the broadening of the absorption band observed at approximately 700 nm is attributed to the two electronic transitions in the *d* orbital corresponding to $^2A_{1g} \rightarrow ^2B_{1g}$ and $^2B_{2g} \rightarrow ^2B_{1g}$ [39]. In general, this means that the bandgap narrows as the longer wavelength is absorbed. In this case, it is known that the excitation of electrons easily occurs even under weak light, thereby increasing the activity of the photocatalyst. Based on the absorption peak, which is the largest among the absorption wavelengths, the bandgaps decrease in the order of $Ni_3(PO_4)_2$-$Ni_2P_2O_7$mpc > $Co_3(PO_4)_2$-$Co_2P_2O_7$mpc > $Cu_3(PO_4)_2$-$Cu_2P_2O_7$mpc. The wavelengths of 430, 580, and 670 nm were calculated for the $Ni_3(PO_4)_2$-$Ni_2P_2O_7$mpc, $Co_3(PO_4)_2$-$Co_2P_2O_7$mpc, and $Cu_3(PO_4)_2$-$Cu_2P_2O_7$mpc samples using the following equation:

$$E_g = hc/\lambda \tag{1}$$

where $E_g$ is the bandgap energy, h is Planck's constant, c is the speed of light, and $\lambda$ is the given wavelength [40]. Bandgaps of 2.88, 2.14, and 1.77 eV are then obtained. However, in the $Cu_3(PO_4)_2$-$Cu_2P_2O_7$mpc sample, the yellow-green color at approximately 410 nm and the dark blue color at 700 nm must be combined to obtain a light-blue colored $Cu_3(PO_4)_2$-$Cu_2P_2O_7$mpc compound, which we synthesized. Thus, we recognize that both absorption peaks must be considered, and the bandgap at 410 nm is approximately 3.0 eV. In the literature, it was confirmed that $Cu_3(PO_4)_2$ absorbs light at wavelengths of 380 nm and 650 nm [41], and $Cu_2P_2O_7$ does so at a wavelength of 700 nm or longer [42].

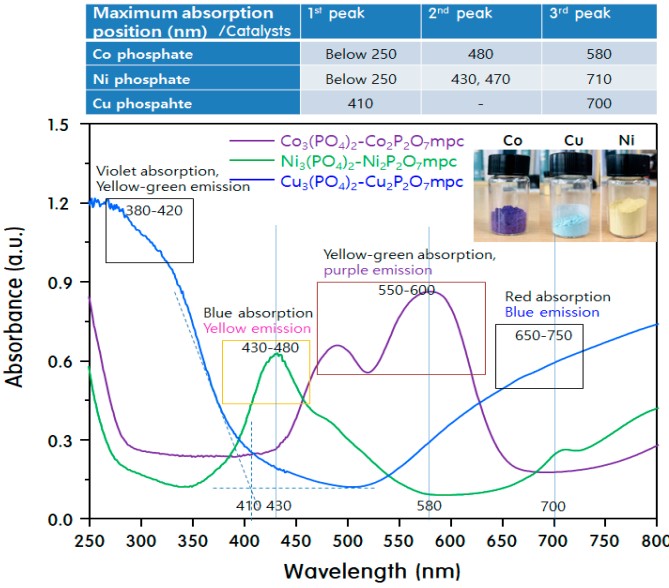

| Maximum absorption position (nm) /Catalysts | 1st peak | 2nd peak | 3rd peak |
|---|---|---|---|
| Co phosphate | Below 250 | 480 | 580 |
| Ni phosphate | Below 250 | 430, 470 | 710 |
| Cu phospahte | 410 | – | 700 |

**Figure 4.** UV-Vis spectra of the $M_3(PO_4)_2$-$M_2P_2O_7$mpc (M = Co, Ni, and Cu).

By contrast, electrons exciting from the valence band (VB) to the conduction band (CB) are recombined with holes, and photoluminescence (PL) is a useful method for predicting the degree of recombination. The PL results for the three samples when the electrons are excited by light with a wavelength of 365 nm are shown in Figure 5. The photoluminescence spectra observed at a 365 nm excitation were found to be dependent on both the structural type and the transition metal ions. The PL spectra of $Co_3(PO_4)_2$-$Co_2P_2O_7$mpc, excited at 365 nm, show only one strong broad emission band centered at approximately 440 nm. It has been fairly accepted that cobalt exists mostly in a divalent state with two coordinations, namely, octahedral and tetrahedral [43]. A $Co^{2+}$ ion has a $d^7$ electronic configuration, and in a tetrahedral crystal field, presents the splitting of energy levels of a $d^3$ electronic configuration in an octahedral field. In octahedral coordination ($Co^{2+}$), the free ion ground state 4F splits into $^4T_1$, $^4T_2$, and $^4A_2$ states with the $^4T_1$ state being the lowest. In a tetrahedral symmetry, the energy levels of $Co^{2+}$ ions are $^4T_2(4F)$, $^4T_1(4F)$, $^2E(2G)$, and $^4T_1(4P)$, with a ground state of $^4A_2(4F)$. The emission band of $Co^{2+}$ ions within the region of 630–670 nm is assigned to $^2E(2G) \rightarrow ^4A_2(4F)$

of $Co^{2+}$ ions in tetrahedral coordination [44]. The PL spectra $Ni_3(PO_4)_2$-$Ni_2P_2O_7$mpc show three strong broad emission bands centered at approximately 450, 465, 510, and 610 nm. $Ni^{2+}$ ions ($3d^8$) are expected to exist as octahedral and tetrahedral coordination sites. The luminescence of $Ni^{2+}$ ion-doped phosphate can be associated with the d–d optical transitions. It was reported [45] that the energy levels of $Ni^{2+}$ ions in an octahedral symmetry are $^3A_{2g}(F) \rightarrow {}^4T_{2g}(F)$, $^3A_{2g}(F) \rightarrow {}^3T_{1g}(F)$, and $^3A_{2g}(F) \rightarrow {}^3T_{1g}(P)$. In addition to these three spin-allowed transitions, a spin-forbidden transition $^3A_{2g}(F) \rightarrow {}^1E_g(D)$ could be observed at 610 nm. Many authors have proposed that the luminescence properties of Ni-doped samples exist in two regions, namely, the green (510 nm) and red (610 nm) regions. Hence, according to the energy levels of $Ni^{2+}$ ion transitions in octahedral sites, the emissions in the green and red regions are assigned to the $^1T_{2g}(D) \rightarrow {}^3A_{2g}(F)$ and $^1T_{2g}(D) \rightarrow {}^3T_{2g}(F)$ transitions. Finally, the PL spectra of $Cu_3(PO_4)_2$-$Cu_2P_2O_7$mpc show two strong emission bands extending from 450 to 510 nm within the visible light range. According to the octahedral crystal field, $Cu^{2+}$ ($3d^9$) loses its degeneracy and splits into $^2E_g$ and $^2T_{2g}$, with $^2E_g$ being the lower level. The luminescence spectra of $Cu_3(PO_4)_2$-$Cu_2P_2O_7$mpc exhibit emission peaks at 450, 465, and 510 nm, which are assigned to $3d^94s \rightarrow 3d^{10}$ triplet transitions in $Cu^{2+}$ ions [46]. In general, the lower the PL intensity, the smaller the number of electrons recombined. Therefore, it is expected that the photoactivity of $Cu_3(PO_4)_2$-$Cu_2P_2O_7$mpc and $Co_3(PO_4)_2$-$Co_2P_2O_7$mpc particles are good; in particular, $Cu_3(PO_4)_2$-$Cu_2P_2O_7$mpc particles with a high interfacial transition and defects have high photocatalytic activity. A recent study described the hetero-phase junction phenomenon in association with crystal defects [47]. That is because positive or negative ion defects are generated in crystals, they act as electron or hole capture sites, slowing the recombination rate and increasing the activity of the photocatalyst [48]. The $M_3(PO_4)_2$-$M_2P_2O_7$mpc catalyst synthesized in this study is expected to facilitate the separation of electrons and holes owing to an internal interfacial charge transfer between the inter-phases.

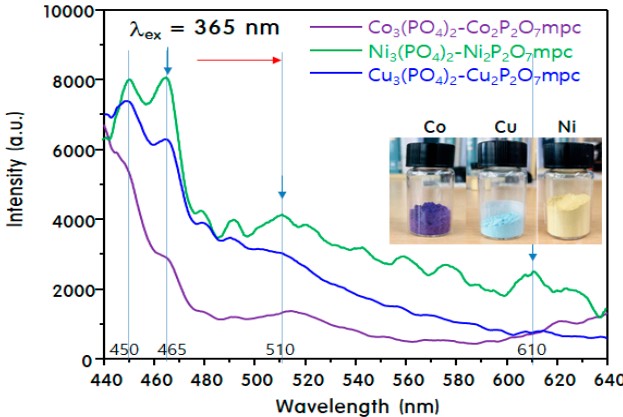

**Figure 5.** PL spectra of the $M_3(PO_4)_2$-$M_2P_2O_7$mpc (M = Co, Ni, and Cu).

In general, effectively separated charges have shown a significant effect on the photoactivity of a catalyst [49]. If the excited electrons flow well on the surface of the catalyst without loosening (that is if the electron and hole are separated well), the reduction reaction in CB and the oxidation reaction in VB will be maintained, and the photocatalytic activity will avoid deterioration. Here, the photocurrent density of the three catalysts was measured, the results of which are shown in Figure 6. For the photocurrent density cycle, the sample to be measured was prepared as a paste and coated onto a cell-type FTO glass. This cell was used as a working electrode, and a Pt-coated FTO glass was used as a counter electrode. The two electrodes were connected to form a single system. An iodolyte electrolyte (AN-50, Solarnonix) was used and sufficiently immersed in the electrode, and a potential of 0 V was applied. The current was measured while illuminating one sunlight of a 2000 solar simulator (IVIUM STAT, ABET Technologies). After the initial 1 min stabilization period, the current was measured repeatedly after the light was applied and removed at intervals of 30 s. The cell area (0.4 cm$^2$) was divided by the measured current, and the current density was finally calculated. The higher the current

density is, the more electrons flowing through the surface of the catalyst without being coupled with the holes. The current density of the $Cu_3(PO_4)_2$-$Cu_2P_2O_7$mpc particles was the largest at 2.75 mA/cm$^2$ after five cycles, followed by $Co_3(PO_4)_2$-$Co_2P_2O_7$mpc and $Ni_3(PO_4)_2$-$Ni_2P_2O_7$mpc particles. This result led us to expect that the photocatalytic activity would be best for $Cu_3(PO_4)_2$-$Cu_2P_2O_7$mpc, which has the highest current density. The results are also consistent with the PL results shown in Figure 5.

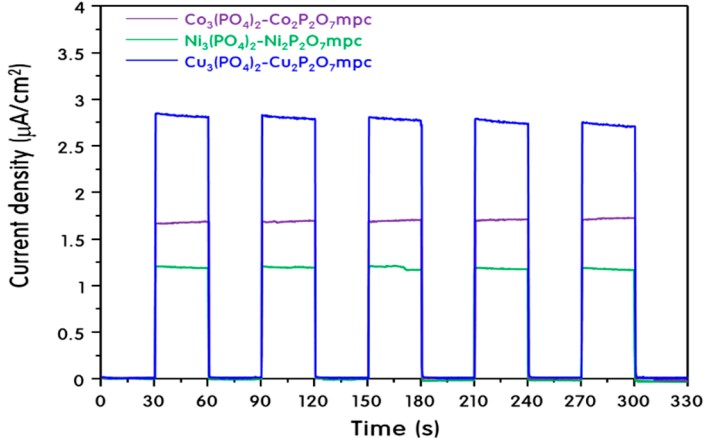

**Figure 6.** Photocurrent responses of the $M_3(PO_4)_2$-$M_2P_2O_7$mpc (M = Co, Ni, and Cu).

Figure 7 shows the hydrogen production achieved by water-splitting under 365 nm light radiation. The amount of hydrogen and oxygen generated gradually increased over time in all catalysts. As expected from the optical properties, a cumulative hydrogen production of 80 μmol/cat.g was obtained after 10 h on the $Cu_3(PO_4)_2$-$Cu_2P_2O_7$mpc, and hydrogen production of 30 μmol/cat.g on the $Co_3(PO_4)_2$-$Co_2P_2O_7$mpc catalyst was obtained. However, $Ni_3(PO_4)_2$-$Ni_2P_2O_7$mpc showed an extremely low hydrogen content of 5 μmol/g, which is within 10% of $Cu_3(PO_4)_2$-$Cu_2P_2O_7$mpc. Copper, in particular, has high reduction potential and strong electron attracting power, which accelerates the movement of electrons. Furthermore, it can easily hydrogenate protons through a redox oxidation-reduction ($Cu^{1+}/Cu^{2+}$) during the reaction. By contrast, the amount of oxygen generated in each sample was exactly half the amount of hydrogen. This result shows that the water decomposition reaction took place quite stoichiometrically on the $M_3(PO_4)_2$-$M_2P_2O_7$mpc catalysts.

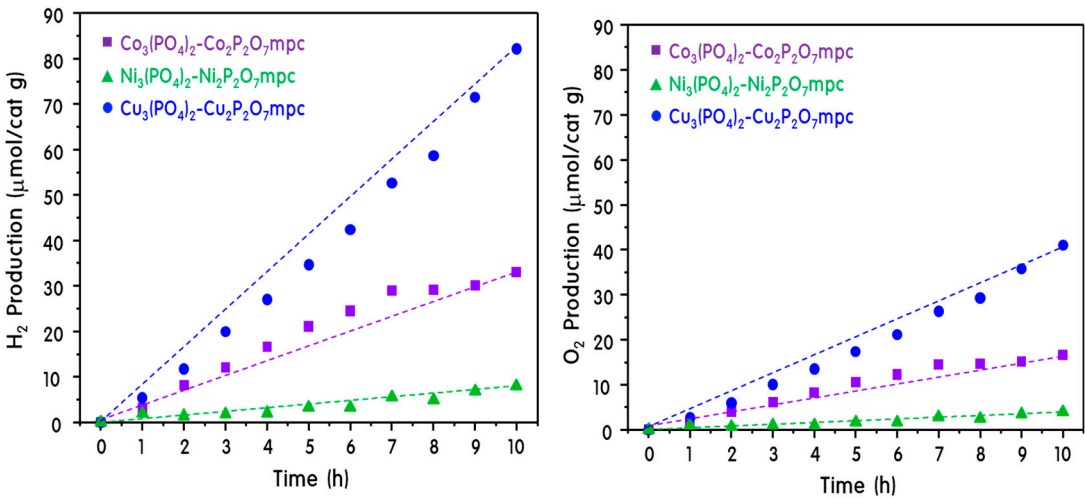

**Figure 7.** Evolutions of $H_2$ and $O_2$ over $M_3(PO_4)_2$-$M_2P_2O_7$mpc (M = Co, Ni, and Cu).

During the water-splitting reaction, the potentials of the CB and VB of the semiconductors are essential to the electron transfer mechanism, and thus, the photocatalytic performance. To establish

an energy potential diagram of $Cu_3(PO_4)_2$-$Cu_2P_2O_7$mpc particles, the VB spectra were obtained from an XPS analysis, the results of which are shown in Figure 8A. The VB of $Cu_3(PO_4)_2$ and $Cu_2P_2O_7$ was found to be 2.48 and 1.20 eV, respectively. Based on the bandgap and VB, the CB was obtained using the following equation [50]:

$$E_{CB} = E_{VB} - E_g. \tag{2}$$

Energy potential diagrams for the $Cu_3(PO_4)_2$ and $Cu_2P_2O_7$ particles are shown in Figure 8B. According to the photocatalytic water-splitting reaction, holes produced in a VB generate oxygen by oxidizing water. In addition, photoelectrons excited in the CB generate hydrogen through water reduction. It is well known that the energy required for a water-splitting reaction is approximately 1.23 eV or more, including both the reduction potential ($H^+/H_2$) and the oxidation potential ($O_2/H_2O$). Based on the results, the $Cu_3(PO_4)_2$ particle satisfies the required energy potential for the water-splitting reaction. It is expected that $Cu_2P_2O_7$ will not cause a water decomposition reaction because the CB value contains the reduction potential of water, whereas the VB value does not include the oxidation potential. However, $Cu_2P_2O_7$ is thought to contribute to a reduction of protons in hydrogen by receiving electrons relaxed from $Cu_3(PO_4)_2$. Furthermore, the internal electron transfer system facilitates the separation of electrons and holes, and thus, their recombination is suppressed, thereby improving the catalytic activity.

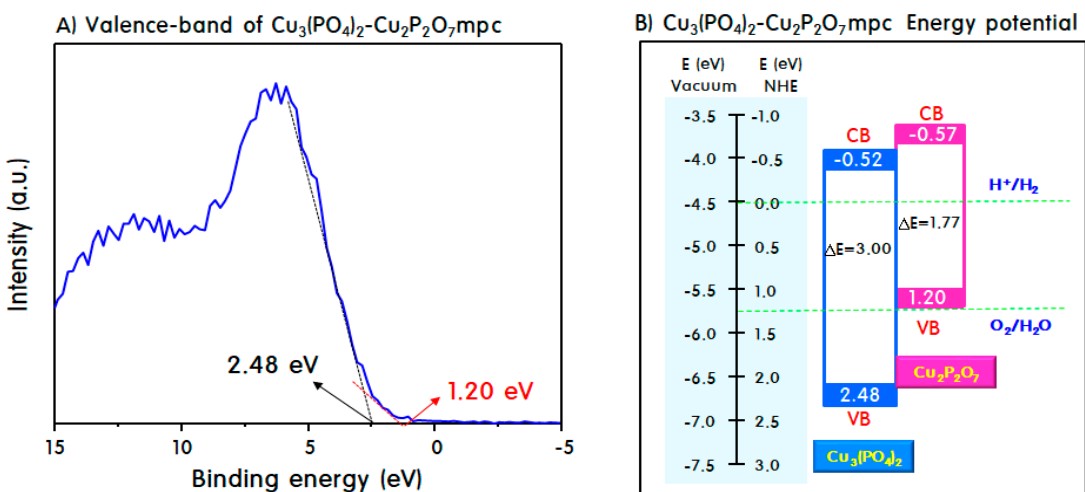

**Figure 8.** (**A**) VB spectra determined from the XPS study of $H_2$ and $O_2$ over $M_3(PO_4)_2$-$M_2P_2O_7$mpc (M = Co, Ni, Cu) and (**B**) electron transfer energy diagram for $M_3(PO_4)_2$-$M_2P_2O_7$mpc (M = Co, Ni, and Cu).

Scheme 1a,b show the expected water-splitting mechanisms based on optical properties and hydrogen production performance when the $Cu_2P_2O_7$ mixed-phase in the $Cu_3(PO_4)_2$ catalyst ($Cu_3(PO_4)_2$-$Cu_2P_2O_7$mpc) was present. According to Figure 8, the conductor band and valence band of $Cu_3(PO_4)_2$ (and $Cu_2P_2O_7$) had been measured to −0.52 eV (−0.57 eV) and 2.48 eV (1.20 eV), respectively. In Scheme 1a, the bandgap position of $Cu_2P_2O_7$ particle does not contain the redox potential for water decomposition, and it has the reduction potential of hydrogen but has not the oxidation potential of oxygen [51]. Eventually, the Scheme 1a mechanism is undesirable in this study, even though the $Cu_3(PO_4)_2$ includes the redox potential for water decomposition [52]. Thus, like Scheme 1b, excited electrons can only move. The Z-scheme mechanism is well-known and refers to when electrons are moved to the Z-type [53]. In particular, when the light of a wavelength of 365 nm is irradiated, the charge transfer can be performed like the Z-scheme system as shown in the red dotted line in Scheme 1b. In particular, when two phases are internally connected rather than heterogeneous, and electrons can move through the internal interfacial transition between these two phases. Furthermore, Wu et al. reported that an interfacial internal electric field is formed in a direct

Z-scheme photocatalyst synthesized by an in situ growth method and the photocatalytic activity is enhanced by an internal charge transfer mechanism [54]. Although their catalysts are somewhat different from the catalysts in this work, it is assumed that almost similar forms of internal interfacial barriers have been formed, and thus, water can be expected to be decomposed by an interfacial charge transfer mechanism in the same context.

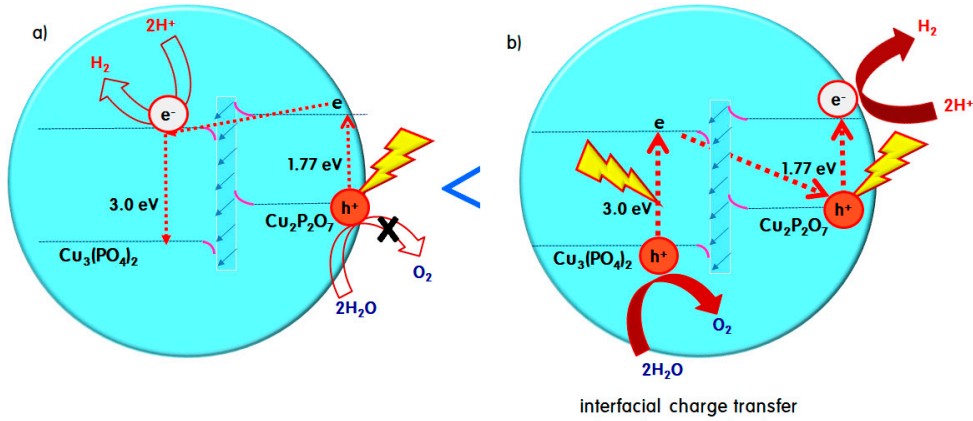

**Scheme 1.** The expected water degradation mechanism when the initial electron excitation occurs in the valence band of (**a**) $Cu_2P_2O_7$ and (**b**) $Cu_3(PO_4)_2$.

## 3. Experimental

### 3.1. Synthesis of Catalysts

The synthesis procedures of the $M_3(PO_4)_2$-$M_2P_2O_7$mpc particles are shown in Figure 9, the specific method of which is as follows: Water in the solvent was placed in an Erlenmeyer flask, and metal nitrates (99.95%, $Co(NO_3)_2 \cdot xH_2O$, $Ni(NO_3)_2 \cdot xH_2O$, $Cu(NO_3)_2 \cdot xH_2O$, Junsei Co., Tokyo, Japan) were quantitatively added and uniformly stirred for 1 h until completely dissolved. Sodium hydrogen phosphate (99%, $Na_2HPO_4$, Junsei Co., Japan) as a phosphate source was added such that the ratio of metal to $PO_4$ was 3:2, followed by stirring for 2 h. The chemical reaction at this time was $MNO_3$ + $Na_2HPO_4 \rightarrow MHPO_4$ + $Na_2NO_3$, and the amorphous $MHPO_4$ was precipitated. The precipitate was washed and then dried at 70 °C for 24 h. The dried powder was sintered in an electric furnace at 700 °C for 4 h under air conditions. At this time, a chemical reaction occurred as $6MHPO_4$ + $3/2O_2 \rightarrow 2M_3(PO_4)_2$ + $3H_2O$, and finally, we obtained the three types of crystallized $Co_3(PO_4)_2$-$Co_2P_2O_7$mpc, $Ni_3(PO_4)_2$-$Ni_2P_2O_7$mpc, and $Cu_3(PO_4)_2$-$Cu_2P_2O_7$mpc particles.

### 3.2. Characterizations

The crystal structures and shapes of the synthesized $Co_3(PO_4)_2$-$Co_2P_2O_7$mpc, $Ni_3(PO_4)_2$-$Ni_2P_2O_7$mpc, and $Cu_3(PO_4)_2$-$Cu_2P_2O_7$mpc particles were identified through XRD (X'Pert Pro MPD PANalytical, nickel-filtered $CuK\alpha$ ($\lambda$ = 1.5406 Å, 30 kV, 15 mA, $2\theta$ angle = 10–80°) and TEM images (H-7600, Hitachi, Tokyo, Japan). A diffuse-reflectance ultraviolet-visible spectrometer (wavelength of 200–800 nm, DIR-UV–Vis, Neosys-2000, Scinco Co., Seoul, Korea), photoluminescence spectroscopy (wavelength of 320 nm, PL, Perkin Elmer, He-Cd laser source), and the photocurrent (using a 2000 solar simulator, ABET Tech., Milford, CT, USA) were used to determine optical properties of the $Co_3(PO_4)_2$, $Ni_3(PO_4)_2$, and $Cu_3(PO_4)_2$ particles.

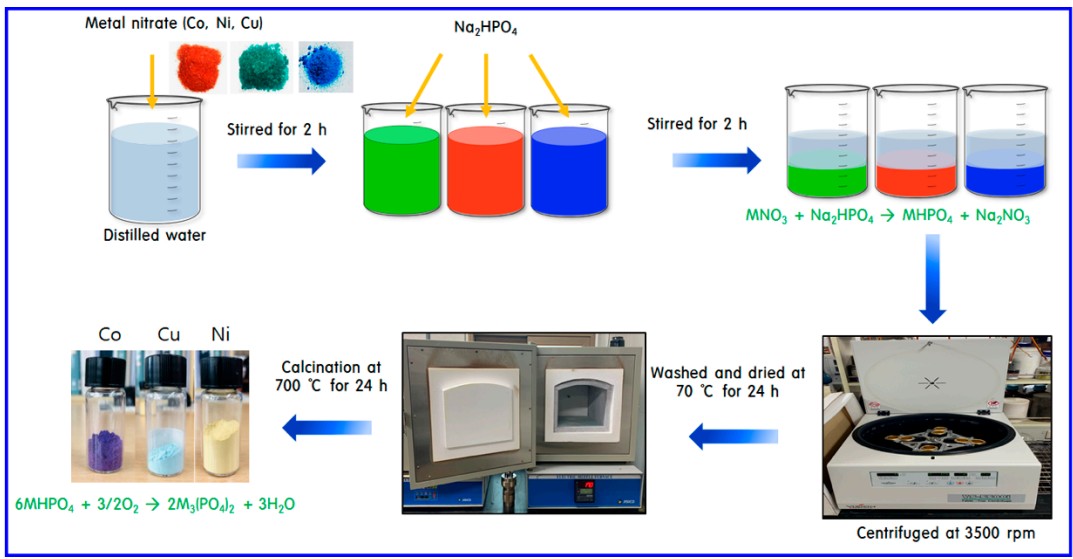

**Figure 9.** Overview on preparation process of the $M_3(PO_4)_2$-$M_2P_2O_7$mpc (M = Co, Ni, and Cu).

### 3.3. Hydrogen Production through Water Photo Splitting

The photocatalytic decomposition of DI water was carried out using a liquid photoreactor prepared in our laboratory, as shown in Figure 10. First, the photocatalytic decomposition of an aqueous solution without scavengers using a UV light source was conducted using a Pyrex reactor. Next, 1.0 L of distilled water was placed into the reactor, and 0.5 g of synthesized $Co_3(PO_4)_2$-$Co_2P_2O_7$mpc, $Ni_3(PO_4)_2$-$Ni_2P_2O_7$mpc, and $Cu_3(PO_4)_2$-$Cu_2P_2O_7$mpc catalyst powder was added. The light was irradiated using a UV lamp ($3 \times 6$ cm$^2$ = 18 W cm$^2$, length of 30 cm, a diameter of 2.0 cm, Shinan, Pochon, Korea) at a wavelength of 365 nm, and the reaction was conducted for a total of 10 h. To investigate the oxidation states of the Cu2p, P2p, and O1s components and the valence band values of the prepared sample, X-ray photoelectron spectroscopy (XPS) (AXIS Nov, Kratos, Inc., San Diego, CA, USA) was used. The resulting gas was analyzed through gas chromatography (GC, DS7200, Donam Co., Gwangju, Korea). For the GC conditions, a thermal conductivity detector and a Carboxen-1000 column (Bruker, Billerica, MA, USA) were applied, along with injection, oven, and detector temperatures of 423, 393 and 473 K, respectively.

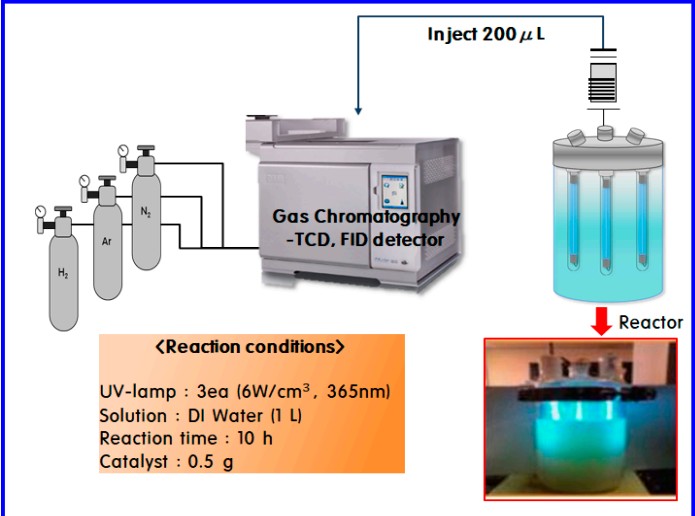

**Figure 10.** A liquid photoreactor designed in our laboratory for the photocatalytic splitting of water.

## 4. Conclusions

This study focused on improving the performance of the catalyst by promoting the charge transfer through the internal interface without heterojunctions between other crystals. Metal (Co, Ni, and Cu) phosphates having $3d^7$, $3d^8$, and $3d^9$ valence electrons were prepared and their photocatalytic activities were compared during a water-splitting reaction. An XRD analysis confirmed that the monoclinic octahedral structured $Co_3(PO_4)_2$, $Ni_3(PO_4)_2$, and $Cu_3(PO_4)_2$ partially co-exist with $Co_2P_2O_7$, $Ni_2P_2O_7$, and $Cu_2P_2O_7$ phases (called $M_3(PO_4)_2$-$M_2P_2O_7$mpc) because their phase equilibrium energies are in similar locations. Absorption peaks for $Co_3(PO_4)_2$-$CO_2P_2O_7$mpc, $Ni_3(PO_4)_2$-$Ni_2P_2O_7$mpc, and $Cu_3(PO_4)_2$-$Cu_2P_2O_7$mpc were observed at various wavelengths depending on the d-d electron transition of the metallic components. The intensity of the PL peak related to the recombination between electrons and holes increased in the order of $Cu_3(PO_4)_2$-$Cu_2P_2O_7$mpc < $Ni_3(PO_4)_2$-$Ni_2P_2O_7$mpc < $Co_3(PO_4)_2$-$CO_2P_2O_7$mpc. The photocurrent density correlated with the charge separation between the electrons and the holes decreased in the order of $Cu_3(PO_4)_2$-$Cu_2P_2O_7$mpc > $Ni_3(PO_4)_2$-$Ni_2P_2O_7$mpc > $Co_3(PO_4)_2$-$CO_2P_2O_7$mpc. The water-splitting performance was found to be the best in $Cu_3(PO_4)_2$-$Cu_2P_2O_7$mpc, and cumulative hydrogen production of 80.0 μmol/g was observed during a 10 h period. This is probably attributed to Cu ions in $Cu_3(PO_4)_2$-$Cu_2P_2O_7$mpc with high reduction potential easily capturing electrons and decomposing $H_2O$ into $H^+$ and $OH^-$. In addition, it was confirmed that the performance of a catalyst can be improved through the effective separation of electrons and holes induced by electrons migrating between the internal interfaces of $M_3(PO_4)_2$-$M_2P_2O_7$mpc.

**Author Contributions:** Conceptualization—M.K.; Data curation, J.K. and J.Y.D.; Formal analysis—J.K. and J.N.H.; Investigation—S.J.Y. and Y.K.; Methodology—S.Y.J., Y.K.; Supervision—M.K.; Writing—original draft—J.Y.D.; Writing—review & editing—M.K.

**Funding:** This work was supported by the National Research Foundation of Korea (NRF) grant funded by the Korean government (MSIT) (No. 2018R1A2B6004746).

**Conflicts of Interest:** The authors declare no conflict of interest.

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
