# Peer review of "Hydrogen Production Improvement on Water Decomposition Through Internal Interfacial Charge Transfer in M3(PO4)2-M2P2O7 Mixed-Phase Catalyst (M = Co, Ni, and Cu)"

_catalysts, doi:10.3390/catal9070602_

Round 1

Reviewer 1 Report

Review for Hydrogen production improvement on water decomposition through internal interfacial charge transfer in M3(PO4)2-M2P2O7 mixed phase catalyst (M=Co, Ni, and Cu)

In the manuscript, the authors compared three different metals-based mix-phase catalysts for hydrogen evolution reaction. The authors synthesized the catalysts, analyzed the catalysts’ performance, and provided a hypothesis to explain the difference of the catalytical activities. Several minor points need to be addressed before publishing in Catalysts.

1.       In the introduction, the authors mentioned TiO2 photocatalytic activity and its limitation. One of them is the relatively large bandgap. The phosphate-based materials may provide the small bandgap which can utilize the visible light for the hydrogen evolution. However, in Figure 7, the authors still used the 365 nm light for the excitation to demonstrate the photocatalytic performance. Furthermore, the absorption spectra of the three catalysts show low absorbance at 365 nm. Why the authors choose 365 nm excitation instead of the blue or green light excitation?

2.       For Figure 4, it could be helpful if the authors list all the electron transitions in the absorption spectra.

3.       Figure 5, the photoluminescence spectra for all three cases are very noisy. Please explain the oscillations of the spectra.

4.       For the photocurrent responses in Figure 6, could authors provide the experimental details, such as electrolyte and applied voltage if applicable.

Author Response

Thank you very much for your kind comments.

Based on your comments and advice, we wrote the answer and revised the manuscript.

Thank you very much.

Reviewer 2 Report

This paper deals with the photocatalytic performance on the water splitting by M3(PO4)2-M2P2O7 catalysts. The topic of the manuscript is interesting and it reports some interesting information on the subject, but in my opinion there are only some points that should be clarified before the publications.

In particular, the authors should take into consideration the following considerations:

- I strongly suggest to re-write the abstract. It should be more clear and attractive. It can not start with “This work….without hetero-junctions between other crystal”. For a new reader it is extremely difficult understand the main topic of the paper, moreover the “catalyst” should be presented before explaining better the peculiarities of these systems. Also the physical-chemical properties should be presented differently,it seems only a list.

-Introduction lines 42-43: “Catalysts…being published”. The authors should add some references. I suggest these recent one (Appl. Catal. B: Environ. 224 (2018) 136–145; International Journal of Hydrogen energy 44 (2019) 14796-14807; Appl. Catal. B: Environ. 236 (2018), 396-403.

-Line 46: The authors should add more detail on the Nasicon-type material as the chemico-physical properties reported in the literature, their utilization also in other fields, ecc. Furthermore, it is important have an idea about the toxicity of these materials. Being them involved in environmental friendly reactions it is crucial have information about this aspect.

Pag. 3 line 82: “The Cu2p3/2… the number 3/2 should be as subscript.

Pag. 5 line 137: “In general, effective…..include “effective…..charge” please re-written the sentence in this form has no sense.

Pag. 6 line 169: It is not clear how the authors have determined the Eg value. Please explain.

Pag. 7 line 190: “These two….homogeneously mixed rather than heterogeneous..” What are the experimental data that induce at this conclusion?”

Pag. 9 paragraph 3.3.: The authors written (also in the conclusion) that the photocatalytic experiments are conducted without the addition of scavengers, i.e. overall water splitting experiments. But in figure 10 it is showed that the solution is methanol:water. Please explain and correct. If the authors used methanol they detected CO2 or other products?

Pag. 10 line 243 the word photcatalysts should be corrected.

-Please revise the English form in all the manuscript.

In my opinion the manuscript can be published in Catalysts after a major revision according to the above reported comments.

Author Response

(The authors gave the same response as above.)

Reviewer 3 Report

In this Article the Authors report the preparation of some metallic phosphate-diphosphate mixed phase compounds, as catalysts for water splitting.

The writing of the manuscript is inadequate, English must be improved, some parts need to be improved, some critical issues need to be clarified. Some suggestions follows:

1.    The introduction should be improved, the state of art is deficient.

2.    The description of the diffractograms is incomplete and unclear. Figure 1 shows a series of unassigned peaks, it is not clear if impurities, phase transitions or other are present. Lines 62-64, the sentence: “Especially, the phase equilibrium position of M2P2O7 is more stable than M3(PO4)2. Therefore, in conclusion, it can be seen that M3(PO4)2 monoclinic structure is difficult to synthesize as a perfect crystal”, should be clarified and possibly rewritten.

3.    Line 67, the sentence: ” The TEM image in Figure 2 also shows the same trend as XRD result”, should be clarified. As written, the discussion on TEM provides no contribution to understanding the characteristics of catalysts. It would be better to show images at 20 nm.

4.    Why XPS was performed only on copper phosphate? Lines 90-91: the sentence: “This result is similar to that of XRD, and it is concluded that the Cu3(PO4)2 was coexisted with Cu2P2O7 phase in Cu3(PO4)2-Cu2P2O7mpc” should be clarified.

5.    Lines 109-111: a bibliographic reference is missing.

6.    Figure 7, what do the dashed lines represent? Trend lines?

7.    The discussion on the mechanism is interesting, however the theories must be demonstrated. The Authors should focusing on the experimental evidence, improving the discussion.

8.    Paragraph 3.1: the description of the preparation procedure should be improved, some sentences are not clear.

9.    Numerous typing errors are present

Author Response

(The authors gave the same response as above.)

Round 2

Reviewer 2 Report

The authors have satisfactory answered to the questions. For my point of view the manuscript can be accepted in the present form.

Author Response

Thank you for your kind comments. I attached a response to your comment.

Reviewer 3 Report

I thank the Authors for having responded to all the issues identified in the first version of the manuscript. The manuscript has been greatly improved, however two critical issue were not completely addressed.

1.    The assignment of the XRD diffractograms is uncompleted.

2.    The Authors stated that the dashed lines, in figure 7 are trend lines, it seems so strange trend. Please provide equation and R2.

Author Response

(The authors gave the same response as above.)

Round 3

Reviewer 3 Report

Dear Authors the definition of trend line is unambiguous, so I don't think to be confused, as you stated in your response. 

Defining the reviewer "confused" is an offense. The intent of the reviewer is always to provide suggestions for improving the Articles. If an author confuses a trend line with a line connecting the starting point to the final point, the reviewer has the duty to warn him. If the authors try to remedy the problem by writing that the reviewer is confused, then their behavior is not adequate

Moreover the trend line corresponding to Cu3(PO4)-Cu2P2O7-mpc seems to have a change in slope at the point 7 (time h), so it can correspond to the equation and R2 reported.

Author Response

I fully agree with your comments.

Your comments have been very helpful in improving our articles.

We acknowledge our mistake in expressing wrong, and express our apologies and gratitude to you.